# *rworkflows*: automating reproducible practices for the R community

Brian M. Schilder [1,2] ✉, Alan E. Murphy [1,2] & Nathan G. Skene [1,2] ✉

Despite calls to improve reproducibility in research, achieving this goal remains elusive even within computational fields. Currently, >50% of R packages are distributed exclusively through GitHub. While the trend towards sharing open-source software has been revolutionary, GitHub does not have any default built-in checks for minimal coding standards or software usability. This makes it difficult to assess the current quality R packages, or to consistently use them over time and across platforms. While GitHub-native solutions are technically possible, they require considerable time and expertise for each developer to write, implement, and maintain. To address this, we develop *rworkflows*; a suite of tools to make robust continuous integration and deployment (https://github.com/neurogenomics/rworkflows). *rworkflows* can be implemented by developers of all skill levels using a one-time R function call which has both sensible defaults and extensive options for customisation. Once implemented, any updates to the GitHub repository automatically trigger parallel workflows that install all software dependencies, run code checks, generate a dedicated documentation website, and deploy a publicly accessible containerised environment. By making the *rworkflows* suite free, automated, and simple to use, we aim to promote widespread adoption of reproducible practices across a continually growing R community.

Reproducibility is essential to the progress of research. Yet, >70% of researchers reported being unable to reproduce previously published results, according to a 2016 survey by *Nature*[1]. There are a variety of reasons contributing to this including pressure to publish, selective reporting, and methods not being reported in sufficient detail to replicate. Due to the programmatic nature of data analysis, there are unique opportunities to systematically maximise reproducibility and methodological transparency in this domain. Despite this, surveys of PubMed and GitHub have revealed that between 68 and 70% of bioinformatics resources were never used beyond the original publication[2,3]. Contributing factors may include a lack of coding standards, changing software dependencies, insufficient documentation, and discontinued maintenance post-publication. While general guidelines have been proposed for making software FAIR (Findable, Accessible, Interoperable and Reusable)[4], exclusively placing the

burden on individual developers to design and implement FAIR solutions is insufficient to stimulate substantial progress in this direction[5]. Instead, providing tools to automate FAIR protocols that can be easily applied to a wide variety of software applications with minimal effort and maximal reward for the individual developer are more likely to receive widespread adoption by the scientific community.

Within the sciences, especially bioinformatics and computational biology, R[6] has become one of the most commonly used programming languages[3,7]. Initiatives such as The Comprehensive R Archive Network (CRAN), Bioconductor (Bioc)[8,9], rOpenSci[10,11], and R-Forge have made great strides towards improving the accessibility and robustness of R packages through establishing centralised repositories that require certain coding/reproducibility standards. There are R functions to check whether a given package meets best-practice coding standards include *rcmdcheck* (for CRAN standards)[12], *BiocCheck* (for Bioc

[1]Department of Brain Sciences, Faculty of Medicine, Imperial College London, London W12 0BZ, UK. [2]UK Dementia Research Institute at Imperial College London, London W12 0BZ, UK. ✉e-mail: brian_schilder@alumni.brown.edu; n.skene@imperial.ac.uk

standards)[13], and *pkgcheck* (for rOpenSci standards)[14]. However, initially learning how to set up R packages such that they are compatible with these standards, and manually rerunning checks to ensure they continue to meet these standards, incur non-trivial costs in terms of both time and effort. Even if all checks pass on one's local machine, this does not guarantee that the same software will run as expected on a different Operating System (OS) (e.g. due to version/availability conflicts across many software dependencies). Most journals, funders, and institutions do not systematically check software for any meaningful quality or reproducibility standards, nor do they check for continued maintenance. It is therefore usually left to each research group to decide how rigorously they test their software, a process which is often opaque to users. Presently, many softwares are exclusively distributed through GitHub, due to the ease of doing so and the perceived challenges of submitting to dedicated R package repositories such as CRAN/Bioc/rOpenSci. Unlike these dedicated R package repositories, GitHub does not require R packages (or any other software) to meet any quality standards, or even install or run. In the absence of additional safeguards, this leaves even more opportunities for such softwares to fail or produce erroneous results. This is problematic for not only developers when assessing the quality and fail points of their own software, but for all stakeholders in the R community, including users, research groups, companies, or any downstream entity that relies on results generated by these software.

A prevalent culture of openly sharing software source code and study-specific analysis scripts on public repositories has undoubtedly helped shift the computational community towards a more transparent, collaborative, and open-source model. Over the last decade, GitHub has rapidly overtaken all other code repositories as by far the most widely used in the fields of bioinformatics and computational biology (>90% in 2017)[3]. In that time, there has been extensive integration of GitHub with other resources such as Zenodo (for example, *rworkflows* Zenodo releases[15]) and Figshare, enabling the assignment of persistent Digital Object Identifiers (DOIs) with public source code (see *Supplementary Information: Links* for more details). At the same time, there have been considerable developments in the scope and depth of tools built directly into the GitHub architecture, including the relatively recent addition of GitHub Actions (GHA). GHA allows any user to run customised Continuous Integration/Deployment (CI/CD) workflows directly on GitHub servers for free and can be triggered simply by pushing updates to one's GitHub repository as usual. These workflows can call upon other bundled scripts hosted elsewhere on GitHub to perform sets of related steps, called "actions". These actions can be triggered to automatically launch by user-selected events, including pushes and pull requests. This ensures that every time a change is made to the underlying code, the software continues to work as expected across multiple OS with a fresh install of all dependencies. However, setting up these workflows currently takes considerable time, effort, and technical expertise.

In an effort to promote FAIRness, as well as enhance software usability and longevity, we developed *rworkflows*: a robust, reusable, flexible and automated CI/CD suite specifically for the development of R packages (Fig. 1). The *rworkflows* suite includes three main components: (1) the *templateR* template: a CRAN/Bioc-compatible R package template that automatically generates essential documentation using package metadata, (2) the *rworkflows* R package: a lightweight CRAN package to automatically setup short, customisable workflows that trigger the *rworkflows* action and (3) the *rworkflows* action: an open-source action available on the GHA Marketplace (see Methods for a more detailed description of each step in the *rworkflows* action). Importantly, the *rworkflows* action is designed to work with any R package out-of-the-box and can be set up by a one-time call to the R function *use_workflow()*. This means users do not need to manually edit any workflow scripts, obviating the need to invest time in learning GHA-specific syntax or configuration. In addition, the *rworkflows*

action produces three main resources. First, a fully containerised installation of the R package and all of its dependencies are automatically created and pushed to a container registry (e.g. GitHub Container Registry, Docker Hub) so that users can easily install local copies of the fully setup environment as either Docker or Singularity containers. Second, it creates a dedicated documentation website entirely from *README* files, in-code *roxygen* notes[16] and vignettes[17], and then deploys the website to the associated GitHub repository via GitHub Pages. Finally, a variety of status reports can be directly displayed in the *README*/landing page of the GitHub repository as badges, such as whether all GHA have been passed, code coverage reports (i.e. what percentage of the total code has been tested), number of downloads, last commit date[18], and more. This allows maintainers and users to immediately assess the current state of the software package.

In an effort to assist the development community in adopting *rworkflows* and make it a *de facto* standard for R package maintainers, we have already begun to expand its user base by making Pull Requests to GitHub repositories of R packages. In particular, we have focused on R packages that have a large user base (e.g. *Seurat*[19,20], *Signac*[21], *ArchR*[22] or are core Bioc dependencies that thousands of other softwares rely upon (e.g. *GenomicRanges*[23], *GenomicFiles*[24], *BSgenome*[25], *rtracklayer*[26], *RSamtools*[27], *VariantAnnotation*[28]). We also present evidence that over 51% of all R packages currently in existence are exclusively distributed via GitHub. This further emphasises the need for robust, GitHub-based quality control/documentation standards that can be frictionlessly utilised by non-experts.

Finally, in collaboration with a multi-national community of developers we have created a step-by-step tutorial guiding users on how to create Bioc R packages using tools including *rworkflows*: https://bioconductor.github.io/bioc_mentorship_docs/bioc-package.html

## Results
### rworkflows adoption
To date, *rworkflows* has been successfully implemented in over 149 R repositories (including forks), and downloaded over 3700 times at an average rate of >300 downloads/month. This includes packages both internal and external to our own research group, as well as the *rworkflows* R package itself. To illustrate this, we created a graph illustrating many of the R packages that currently use *rworkflows*, or depend on packages that do (i.e. second-order dependents) (Fig. 2). As a proxy of *rworkflows*'s downstream impact on the R development community, metadata was systematically gathered from GitHub. Totals across 58 dependents there were: 3089 stars, 758 forks, and 5,0482 downloads (across all distribution repositories).

An interactive and periodically updated version of this graph on the dedicated *rworkflows* documentation website (see Data availability section). This online version also displays the metadata for each repository when users hover the cursor over the respective node.

### GitHub as a package distributor
Most developers who distribute their R packages through dedicated repositories like CRAN, Bioc or rOpenSci still maintain a copy of their software on GitHub for the purposes of development, collaboration and transparency. However many packages go through a lengthy period of development (months to years) before being eventually accepted to one of the dedicated R package repositories. In fact, many developers may never submit their packages to these dedicated repositories, and depending on where and if they publish their work, these packages can be introduced into the scientific community without ever being thoroughly tested. As more software becomes exclusively distributed on GitHub, there is an increased need for GitHub-native solutions which make CI/CD seamless. Since there are currently few to no set standards imposed by journals or GitHub, it is incumbent upon the R developer community to provide tools which

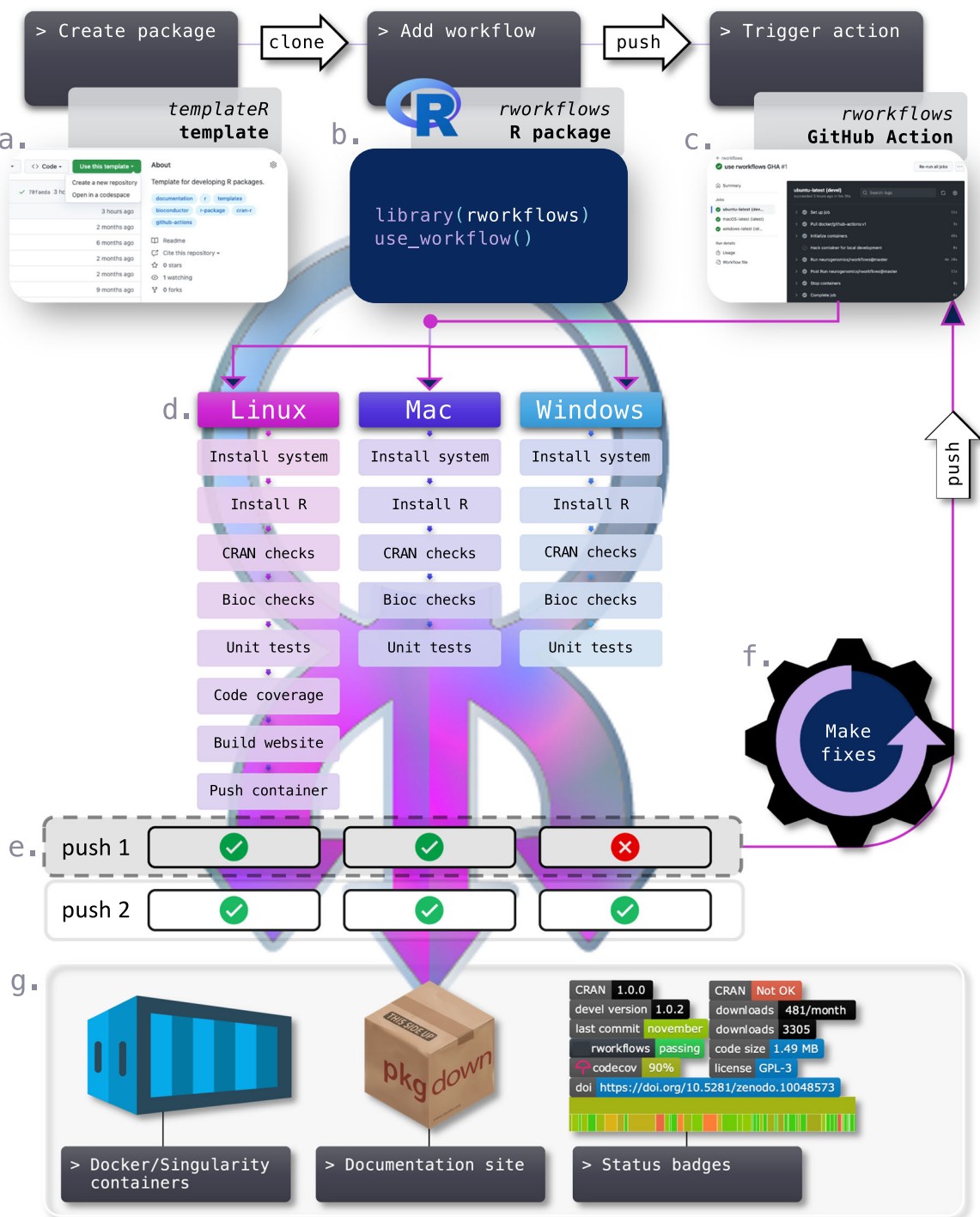

**Fig. 1 | The *rworkflows* suite.** Example usage of *rworkflows*. **a** Create package: create a new R package by forking and cloning the *templateR* template, or use an existing R package. **b** Add workflow: Install the rworkflows R package and use the *use_workflows()* command to generate a workflow *yaml* file in the correct folder structure. Arguments to customise the workflow are detailed in the documentation website. **c** Trigger action: trigger the rworkflows GitHub Action by pushing to GitHub. **d** Run the R package through the workflow on three different OS platforms in parallel. **e** Inspect the results of the workflow run. If one or more workflows fail, an email is automatically sent to the user. **f** If issues are found, make fixes to the software and push again to retrigger the *rworkflows* action. **g** When all workflows have passed, the documentation website is built using *pkgdown*[17] and deployed via GitHub Pages. The containerised R package is then deployed to Docker Hub. Badges embedded into markdown or HTML files (e.g. *README* documentation) will also be automatically updated to reflect the R package's current status. In this figure, a version of the "R" logo with modified colours is used under the terms of the Creative Commons Attribution-ShareAlike 4.0 International license (CC-BY-SA 4.0) (https://creativecommons.org/licenses/by-sa/4.0/). The "container" logo was created by Pause08 and is freely available for reuse via Flaticon (https://www.flaticon.com/free-icon/container_860142). The Codecov logo is used with permission from Codecov. *pkgdown* is provided under an MIT license (https://github.com/r-lib/pkgdown/blob/main/LICENSE.md).

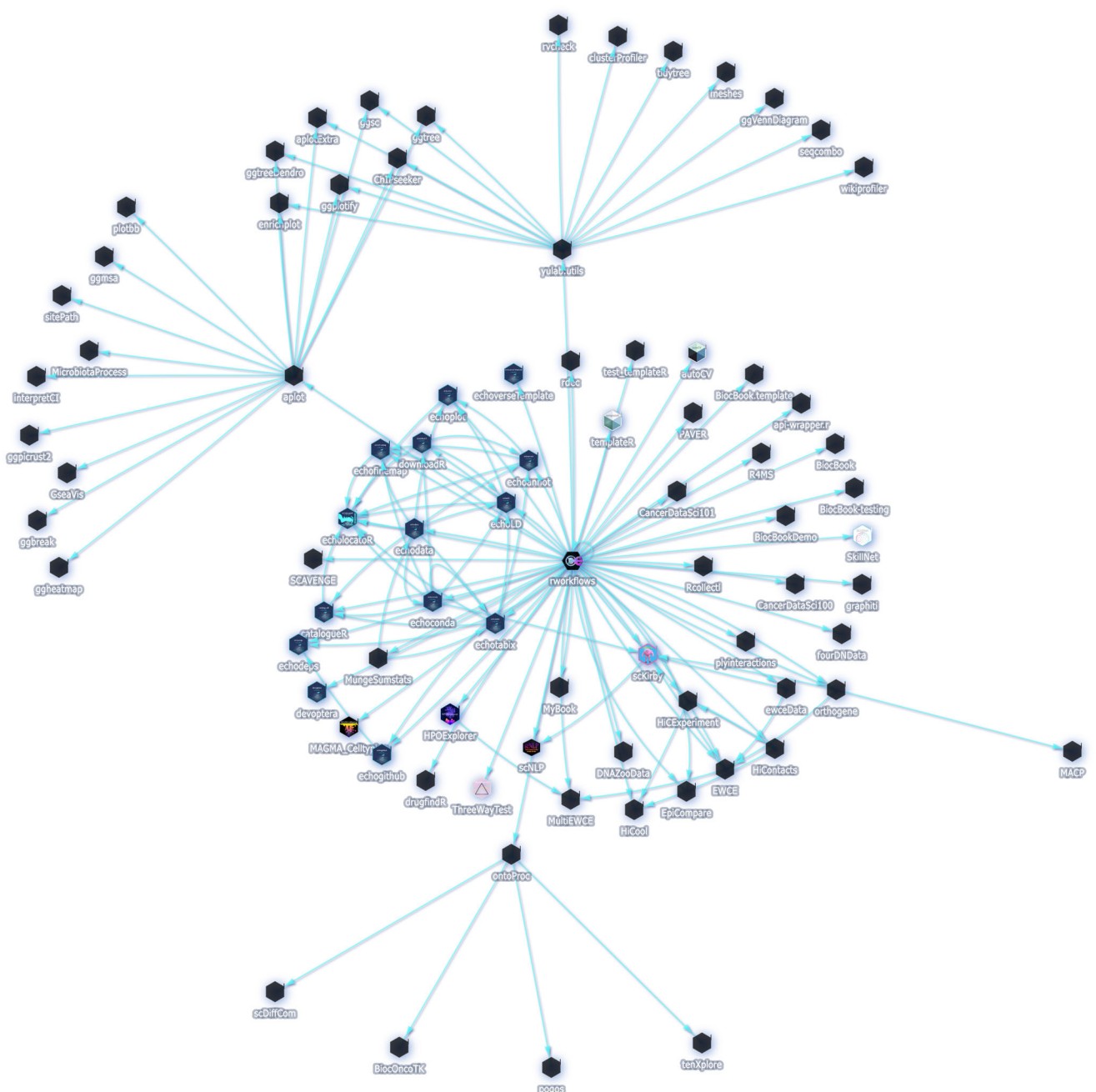

**Fig. 2 | Reverse dependency graph.** A reverse dependency graph showing all R package GitHub repositories (blue nodes) that currently utilise the *rworkflows* action (first-order dependents) or depend on a package that does (second-order dependents). All data was captured from October 24, 2023. An interactive, periodically updated version of this graph is also available online (see Data availability section).

not only make best-practice coding, documentation and CI/CD easy to implement, but immediately beneficial enough to incentivise developers to widely adopt these practices.

To evaluate the magnitude of need for GitHub-based solutions in the R community, we gathered comprehensive data on where repositories R packages are hosted (Fig. 3). An upset plot was generated to visualise how many R packages are distributed via one or multiple repositories. Of the 50,685 R packages we identified, 39.3% (19,932) are available via CRAN, 6.9% (3515) are available via Bioc, 0.63% (318) are available via rOpenSci, 4.3% (2176) are available via R-Forge, and 62.3% (31,592) are available on GitHub. Of particular note, 51% (25,883) of all R packages are exclusively distributed through GitHub. This is likely a

very conservative underestimate, as the data on GitHub R packages comes from a static snapshot previously collected in February 2018, whereas all the CRAN/Bioc/rOpenSci/R-Forge data is fully up-to-date. Thus, over half of all R packages are currently not vetted by dedicated R package distributors and are instead left to the developers to determine their own standards and strategies for reproducibility.

**Comparisons with usethis/biocthis**

It should be noted that there have been at least two other efforts to implement reproducible workflows for R package development via GHA, namely the R packages *usethis*[29] and its Bioc-oriented derivative *biocthis*[30]. While *rworkflows* was heavily influenced by *these packages*,

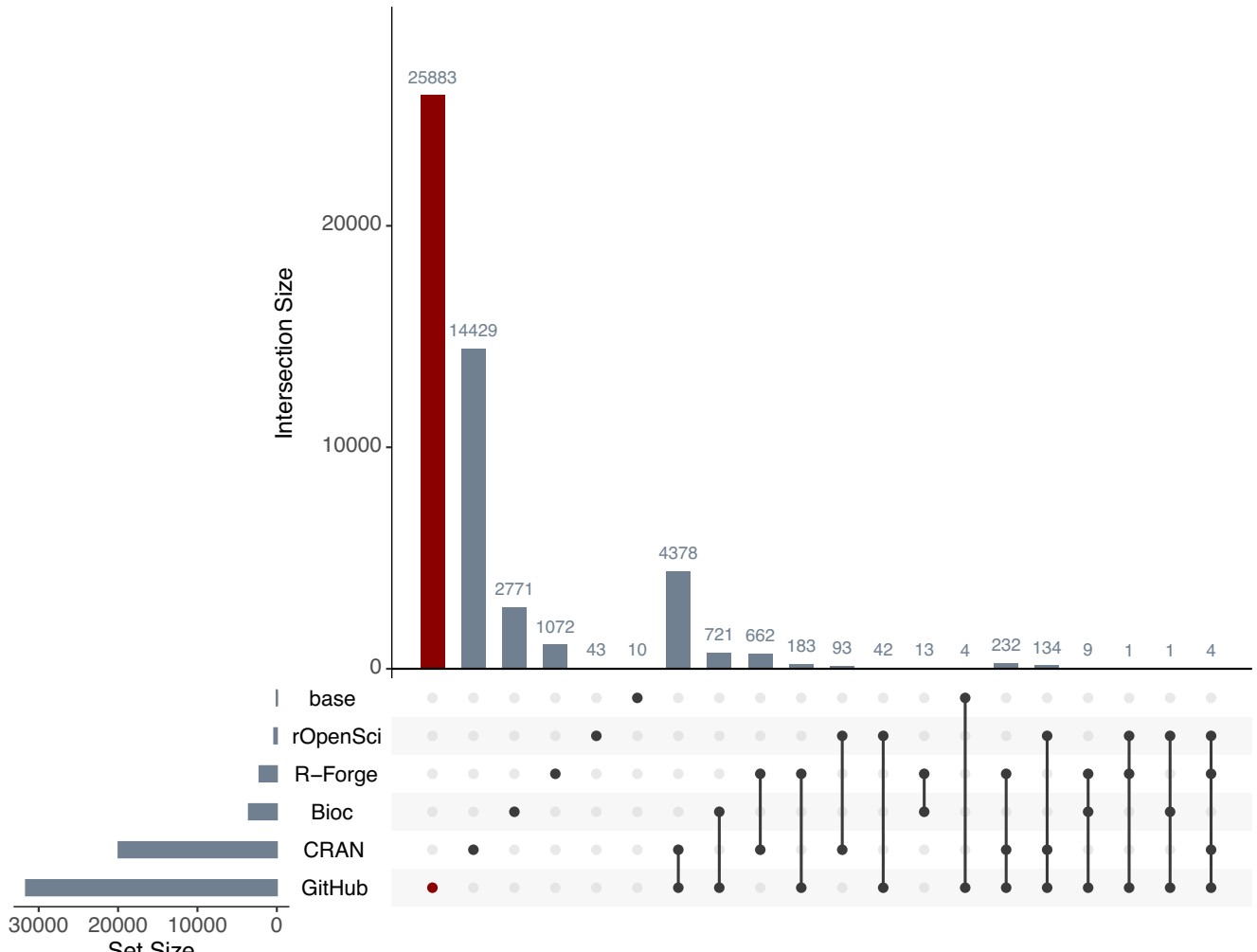

**Fig. 3 | Repositories through which R packages are distributed.** Upset plot of how R packages are distributed through base R, dedicated R packages repositories (CRAN, Bioc, rOpenSci, R-Forge), or code repositories (GitHub). Rows indicate the total number of R packages available through a given distributor. Columns with single dots indicate the number of R packages that are exclusively available through one repository. Columns with multiple dots indicate the number of R packages available via two or more repositories. The number of R packages exclusively distributed through GitHub is highlighted in red.

there are several key differences. First, *rworkflows* operates primarily as an action which is merely called upon by a short workflow script that supplies certain parameters, whereas both *usethis* and *biocthis can only* generate static workflow scripts that dictate each step of the workflow in the file itself. This distinction becomes important when updates need to be made (e.g. new system dependencies, changes to R function implementations, deprecation of certain subactions). Actions such as *rworkflows* need only be updated on the centralised Github repository (see Code availability section), which then propagates to all users who call the *rworkflows* action, even if they implemented *rworkflows* in their package prior to the changes. In contrast, static workflow scripts must be updated by every user individually, and repeated for each GitHub repository. This issue compounds on itself when tasks are split into multiple workflow scripts, as is the case for *usethis*. In some cases, it may take a while for users to infer that the errors they're experiencing are due to changes in the VM provided by the GHA server (for example), rather than something the user is doing wrong, or eventually abandon using the workflow entirely. That said, if users wish to create a more customised workflow that diverges from the *rworkflows* action (and only use it as an initial basis for their script), a full workflow version can be created with *rworkflows::use_workflow(name="rworkflows_static")*, which offers functionality analogous to that of *biocthis*.

Second, *users* can easily control which version of the *rworkflows* action to use with the *tag* argument to indicate a branch (e.g. "master" for the latest version) or release tag (e.g. "v1" for a stable release version tied to a specific commit). Workflow-based strategies like *usethis*/*biocthis* do not enable users to to use different versions of the same workflow, unless they reinstall a different release of the package. In the case of *biocthis*, users must also reinstall all other Bioc packages each time they want to use a different version of the workflow due to Bioc's strongly enforced version control standards.

Third, *rworkflows* offers greater customisability via over 35 fully documented arguments (see documentation website for details: https://neurogenomics.github.io/rworkflows/reference/use_workflow.html) that can be supplied to the *use_workflow* function (with sensible defaults that work out-of-the-box). For example, these arguments allow users to easily choose trigger branches, trigger events, runner OS, code check types as well as the option to support *act*[31], a separate software for running and troubleshooting actions locally before launching them to GitHub. In comparison, the analogous *biocthis::use_bioc_github_action()* function currently has 7 arguments and more limited customisability (*biocthis* v1.12.0).

Fourth, *rworkflows* currently has 92% code coverage via unit tests, whereas *biocthis* (v1.12.0) and *usethis* (v2.2.2) currently have 80% and 57% code coverage, respectively (though this could improve in the

future). Having high code coverage helps to improve package robustness and reduces the chances that the code will break in various use cases[32–35].

Fifth, *rworkflows* obviates the need for a user-supplied Dockerfile as it creates one on the fly instead (see section Container usage). This level of abstraction serves to expand the usage of containers to those who do not know how to successfully set them up manually, or are unfamiliar with the Docker-specific syntax necessary to do so. None of this is to say that the *biocthis* package is obsolete, but rather that it offers other complementary features such as more fine-grained control over template creation than the all-in-one strategy adopted by *templateR*, as well as automated code styling.

Finally, unlike static workflows, all the repositories in which centralised GitHub actions (e.g. *rworkflows*) have been implemented are automatically recorded by GitHub. These can be accessed under the Insights tab of the action's GitHub repository, providing greater insight into the scope of usage and impact of the action (see Fig. 2).

### Comparisons with Bioconductor servers

The *rworkflows* suite is not mutually exclusive to the package checking services provided by Bioc, which regularly run standardised checks on multiple OS. To the contrary, *rworkflows* fills an important gap for developers of Bioc packages who wish to comprehensively test their package before pushing to the upstream Bioc copy, as the upstream copy can take several days to rerun checks. Having an intermediate checking solution via GitHub provides feedback within minutes or hours, as opposed to days, thus greatly accelerating the development cycle. While Bioc does provide a dedicated Docker container with several prerequisite software installed (e.g. *BiocManager*, *BiocCheck*), these containers do not have any other Bioc packages installed. In fact, by default *rworkflows* uses the Bioc Docker container as a base and then builds upon it to generate a package-specific containerised environment ready for distribution to users. This greatly speeds up the time it takes for any given user to successfully install and start using the developer's R package.

### Use case: MAGMA.Celltyping

To demonstrate how *rworkflows* can be helpful in practice, we use the R package *MAGMA.Celltyping* (developed and maintained by our lab) as an example[36]. Since first implementing *rworkflows*, it has revealed a number of vulnerabilities, missing documentation, and bugs within *MAGMA.Celltyping* (https://github.com/neurogenomics/MAGMA_Celltyping/issues?q=). Some of these bugs were only visible when run within a particular version of R (e.g. development) or on a particular OS type. For example, the way we constructed file paths was not robust on Windows OS, and would lead to the software being unable to find key resources on that platform (https://github.com/neurogenomics/MAGMA_Celltyping/issues/92). As none of the developers use Windows machines, this would have been left for users to discover these bugs and (hopefully) report them. In the meantime, some users may have abandoned using our tool without our knowledge. Moreover, running code coverage tests has enabled us to identify potential weak points in our code and design tests that are capable of better assessing these. Whenever we make changes to our code, the coverage badge in the README automatically updates so that we (and our users) know how robust we can expect our tool to be (currently at 75% coverage in v2.0.11, with plans to improve this further). Finally, users of *MAGMA.Celltyping* can now bypass all installation and dependency issues with containers automatically generated by *rworkflows*. The instructions for setting up Docker/Singularity containers were also automatically generated by *rworkflows* (*rworkflows::use_vignette_docker()*), wherein the process of setting up *MAGMA.Celltyping* on any computing environment is reduced to a single copy-and-paste step (https://neurogenomics.github.io/MAGMA_Celltyping/articles/docker).

## Discussion

Most developers would agree that the FAIR principles are noble goals worthy of striving towards. However, the costs associated with putting these principles into practice (e.g. time, learning curves, lack of computational resources) often deter developers from ever effectively implementing them. Therefore, there is a dire need to reduce the burden put on individual developers by automating reproducibe practices, while at the same time increasing the amount of useful output generated by such practices. This will greatly improve the overall cost/benefit ratio of conducting reproducible science, which will in turn incentivise widespread adoption of FAIR and open practices. *rworkflows* aims to do exactly this, by enabling greatly simplified implementation of a robust GitHub-native testing, documentation, and containerisation pipeline through a single R function. This makes *rworkflows* usable by even novice programmers and requires exceedingly minimal local computing power. Furthermore, *rworkflows* can be used in either public or private repositories, extending its utility to pre-production or intellectual property-sensitive packages.

There are no doubt hundreds of invaluable and high-quality R packages that are not hosted on CRAN/Bioc/rOpenSci. Nevertheless, assessing package usability is currently a process of trial and error, which amounts to a huge number of wasted hours compounded across thousands of users. *rworkflows* attempts to make these distinctions more visible and immediately accessible, all while helping to make all R packages meet a set of minimum standards (or at least transparently advertise that they dont yet).

Peer-reviewed journals, as well as repositories like CRAN, Bioc, and rOpenSci, rely almost entirely on volunteer community members to review and approve software packages for official release[11,37]. Each additional cycle in the review–response process due to common and avoidable issues can incur substantial and unnecessary delay. This is only exacerbated by the limited time and considerable demands both parties are faced with[38,39]. *rworkflows* serves to significantly reduce the burden of back-and-forth troubleshooting by decreasing the prevalence of installation errors (through containerisation), coding bugs (through package checks), and miscommunications (through documentation). As the exponentially expanding scientific literature continues to outpace the proportion of qualified researchers willing to volunteer as reviewers[39], making this process more efficient will become increasingly critical for the sustainability of timely, high-quality peer-reviewed research[37,40]. Therefore, journals may wish to consider requiring tools such as *rworkflows* to be implemented as a prerequisite for progressing the review process.

Code coverage is one particularly useful metric for assessing package robustness. While the precise measurement of code coverage varies slightly from one implementation to another, it can generally be summarised as the percentage of lines in your software's code that are run during unit tests (e.g. using the *testthat* or *RUnit* frameworks). This takes into account that code within conditional statements may not be run in all scenarios, and thus encourages developers to test the same code using multiple sets of parameters. Assuming that the tests themselves are valid, a code coverage of 92% could be interpreted as "92% of its code has been systematically tested and is working as intended". Thus, code coverage can serve as a useful, continuous measure of package robustness as it reduces the chances that the code will break in various use cases[32–35]. The *rworkflows* action automatically runs code coverage tests via *covr*[41] and uploads a report to the browser-based Codecov or Coverall services where users can interactively explore which portions of their code are currently not being thoroughly tested. Finally, the *rworkflows::use_badges* function (which builds upon the *badger* package) allows developers to easily advertise both discrete metrics (passing on CRAN/Bioc, passing *rworkflows* checks) and continuous metrics (code coverage percentage) on their GitHub landing pages.

Providing containerised environments with all necessary dependencies pre-installed and an interactive development platform (i.e. *RStudio*[42]) eliminates virtually all installation troubleshooting. This also helps reduce the burden of maintaining software across hundreds to thousands of users, each with one or more slightly different computing environments. As an additional incentive to developers, continued maintenance of bioinformatics tools post-publication is associated with multiple metrics of impact including increased citations[3]. *rworkflows* also allows users to control which versions of R, Bioc, and Python they wish to have installed within the container. By default, it uses the most up-to-date development versions of R/Bioc so that developers can stay ahead of the curve and identify issues in future versions before they have been released to the public. This is important, as it prevents situations where developers are suddenly faced with many bugs that are already affecting a large number of users and must be fixed urgently.

Beyond the initial publication of an R package, *rworkflows* offers a variety of benefits for different stakeholders. Automating clean and consistent documentation website generation without any additional effort encourages developers to keep their documentation up to date and accessible. Having thorough documentation is not only an invaluable resource for new users, but also trainees in the developers' own lab, or even when reteaching themselves after a long period of not being active on the project.

In that same vein, we recognise the importance of ensuring *rworkflows* itself is maintained and extended well into the foreseeable future. We are committed to securing this vision, out of a desire to make this resource continually available to the R community as well as our reliance on *rworkflows* for all of our R packages. It is for this very reason that the *rworkflows* action was designed to, whenever possible, use subactions from well-established developer organisations (e.g. GitHub, R Consortium, Posit, Bioc) as they have the highest likelihood of being maintained in the long term. In addition, we have already begun to explore multiple avenues towards open-source longevity including (but not limited to) seeking official support/collaboration with software repositories, user contributions, crowd-funding, corporate sponsorship, and grants for sustainable software development. All members of the community are encouraged to voice their ideas/concerns/opinions by participating in the dedicated "Longevity" Discussion board (https://github.com/neurogenomics/rworkflows/discussions). In any case, we are dedicated to ensuring *rworkflows* remains open source, well maintained, and free. Additional features already in development include interactive debugging within the GitHub-hosted *rworkflows* action environment, maximising VM storage capacity for resource-intensive R packages, and improved parameter flexibility throughout.

To conclude, the *rworkflows* suite offers an essential toolkit for developers and users of any experience level. This includes developers who (1) currently (or plan to) distribute their R packages through repositories like CRAN/Bioc/rOpenSci and want to run quality checks before resubmitting a new version for official release, (2) wish to exclusively distribute their code through GitHub while maintaining a high level of coding standards, (3) want to keep the documentation updated without constant manual upkeep of a website and/or (4) want to distribute their software in a fully reproducible Docker/Singularity container. Furthermore, the *rworkflows* action is designed to be both easy to use and flexible (through customisable parameters), thus enabling developers to utilise it in whatever way best suits their project-specific goals In practice, this can range from checking for basic installability/usability of an R package, all the way to extensive evaluation of consortia-specific coding and documentation standards with fully automated container deployment. Therefore, *rworkflows* fills a gap that an increasing number of R developers find themselves in by reducing the burden of effectively implementing FAIR practices, and increasing its immediate benefits for developers and users alike.

Finally, to further expand its accessibility we have provided a series of YouTube videos walking new users through the theory and practice of *rworkflows* (https://youtube.com/@NeurogenomicsLab).

## Methods

### templateR template

For users who are creating a new R package from scratch, we have provided a CRAN/Bioc-compatible template (*templateR*). To get started, one simply forks the template by navigating to the GitHub repository (see Code availability section), clicking "Use this template", and cloning a copy of the new R package to begin editing it (Fig. 1a). The user need only replace key metadata fields (e.g. Package, Title, Description, URL) in the *DESCRIPTION* file (a required file for all R packages). What makes this template unique is that all other components of the package (README, vignettes, unit test setup scripts) are programmatically autofilled based on the *DESCRIPTION* file. This strategy greatly minimises redundant and error-prone aspects of R package documentation.

Alternatively, users can start with any pre-existing R package and skip directly to the next step: using *rworkflows* R package. In either case, we have created a companion Wiki page to help guide users who are unfamiliar with the Bioc standards and offer a variety of tips and tricks to make this process easier, which we continue to maintain (see Code availability section).

### rworkflows R package

The *rworkflows* R package is available on both CRAN and GitHub (see Code availability). Workflow scripts (written in *yaml* format) placed within a specific subdirectory within the GitHub repository (.github/workflows/*.yml), dictate which actions are triggered under which conditions. For those not familiar with creating GHA workflows, learning the GHA-specific expressions and idiosyncrasies can be a time-consuming and iterative process. Instead, we have abstracted this step away by autogenerating workflow scripts from a single R command in the dedicated R package: *use_workflow()*. This creates a fully functional workflow file in the correct subdirectory even with no arguments supplied, and only needs to be run once per R package (Fig. 1b). For greater flexibility, users can supply the function with their preferred arguments to generate (or regenerate) a customised workflow script to trigger the *rworkflows* action. By default, the workflow will trigger the *rworkflows* action (see *rworkflows* action section below) upon pushes or pull requests to the remote GitHub repository. For minor pushes (e.g. fixing a typo in the *README* text), one can avoid triggering the action by simply adding the string "[skip ci]" to the commit message. Triggers can be set to activate for specific GitHub branches only (e.g. "main", "master", "devel") or even *regex* expressions (e.g. "RELEASE_**"), which can be quite helpful for developing Bioc packages with regular release updates without having to modify the workflow script each time. Users can even write multiple workflows to the same repository, setting each to trigger via different branches and/or with different parameters (e.g. use the RELEASE_3_18 version of the Bioc docker container when pushing to the RELEASE_3_18 branch). For step-by-step instructions we provide a vignette specifically geared towards Bioc developers (https://neurogenomics.github.io/rworkflows/articles/bioconductor.html). Finally, the *use_workflow()* allows users to control exactly which specific release of the *rworkflows* action they wish to trigger (via the *tag* argument). For a full description of all arguments of the *use_workflow()* function, please refer to the documentation website (https://neurogenomics.github.io/rworkflows/reference/use_workflow.html).

In addition, the *rworkflows* R package contains other useful functions for developers, including *use_badges()*, which dynamically generates badges indicating various aspects of the software package's status to the documentation pages (e.g. the *README* file). It also provides the function *use_dockerfile()*, which writes a Docker recipe file

(i.e. Dockerfile) to create a Docker image with the user's R package (and all of its dependencies) pre-installed). Note that this same function is called automatically in step 8 of the *rworkflows* action, but if a pre-existing Dockerfile in the current working directory is detected, this step is skipped and the pre-existing Dockerfile is used instead. Thus, if preferred, users can have more customised control over how their Docker container is configured. Finally, *use_readme()*, *use_vignette_docker()* and *use_vignette_getstarted()* can generate auto-filed templates for each of these R package documentation components respectively.

## rworkflows action

Once triggered by a workflow, the *rworkflows* action launches three virtual machines (VMs) in parallel to test the R package across multiple OS, including Linux, Mac, and Windows. Within each VM, the following steps are performed (Fig. 1d):

1. Install system: Installs all OS-specific system dependencies that account for a variety of different functionalities that R users may require.
2. Install R: Installs all R dependencies for the R package being tested. Three rounds dependency installation are attempted using slightly different methods to ensure robustness of this procedure without requiring the user to manually troubleshoot this step.
3. Install LaTeX: Install a specific version of LaTeX and any extra LaTeX packages (controlled by the arguments *has_latex*, *tinytex_installer*, *tinytex_version*, and *pandoc_version*).
4. Install conda: Install *conda*, *miniconda*, miniforge, or *mamba* (controlled by the arguments *miniforge_variant*, and *miniforge_version*). Users can additionally provide a yaml file (via the *environment_file* argument) with specifications for a *conda* environment to build and/or activate before all other downstream code checks. This greatly simplifies the installation of not only python packages (which some R packages may use as a backend) but also various extra tools and system dependencies not installed in previous steps. Finally, rworkflows provide a helper R function, *construct_conda_yml()*, for creating new *conda* yaml files for those who are unfamiliar with the formatting requirements.
5. CRAN checks: Run CRAN checks via *rcmdcheck()*. When *run_rcmdcheck = TRUE*, all checks must pass in order for the GHA to succeed. This step uses CRAN standards by default, but can run *rcmdcheck* without CRAN standards by setting the argument *as_cran = FALSE*.
6. Bioc checks: Run Bioc checks via *BiocCheck()*. When *run_bioc = TRUE*, all checks must pass in order for the action to succeed.
7. Unit tests: Runs unit tests implemented via the *testthat*[43] and/or *RUnit*[44] R packages and generates a downloadable report of the results.
8. Code coverage: Runs code coverage tests and uploads the results to Codecov.
9. Build website: (Re)builds the documentation website from *README* files, in-line *roxygen* notes, and vignettes using the *pkgdown*[17]. It then deploys the website via GitHub Pages in a new branch named "gh-pages" in the same repository. Deploying the website via a separate branch is advantageous as it avoids accidentally adding large HTML/CSS/JavaScript source files and libraries to the R package itself (which can slow down its installation and performance in some situations).
10. Push container: Pushes a container to a container registry with your R package, all of its dependencies, and an interactive Rstudio interface pre-installed. Included in *templateR* is an auto-filled vignette for how to create a local Docker or SIngularity container. If you've selected a non-default container registry (e.g. Docker Hub), this step requires a valid authentication token from the relevant registry, which can be stored as a GitHub Secrets variable.

This ensures that only users with appropriate push permissions to a given registry account can update the container there.

Steps 6-8 are only run on the Linux VM to avoid redundancy and avoid conflicts due to simultaneous pushes to their respective repositories (i.e. Codecov, GitHub, Docker Hub).

## Container usage

Containerisation is especially useful when distributing R packages to many users using a wide variety of OS platforms, including high-performance computing (HPC) clusters which may have software installation restrictions for non-root users. Once the *rworkflows* action has successfully completed at least once on the Linux VM, both developers can create Docker and/or Singularity images from the container hosted on a container registry. By default, rworkflows pushes to the GitHub Container Registry, which has the advantage of not requiring any additional accounts or credentials and automatically appearing directly on the associated GitHub repository landing page (under the section "Packages"). Alternatively, users may specify any preferred container registry (e.g. Docker Hub) using the *docker_registry* argument.

If *templateR* was used as a template, a vignette detailing a step-by-step reproducible example is autogenerated. A rendered version of this vignette can be accessed via the dedicated GitHub Pages site, and a link to this vignette is automatically rendered within the *templateR* template *README* file (see Code availability section) under the "Documentation → Docker/Singularity" subheader.

## rworkflows adoption

Metadata was gathered from the GitHub application programming interface (API) for each repository using the R packages *echodeps*[45]. This was used to both identify which packages are currently using the *rworkflows* action (i.e. dependents), and to gather relevant metadata on each of the repositories. Of particular interest were the following metrics; stars (the number of users that bookmarked the GitHub repo with a star), unique clones (the number of unique instances that the GitHub repo was downloaded from Github), and unique views (the number unique instances the GitHub repo was viewed in a web browser). Here, "unique" means the number of distinct internet protocol (IP) addresses. Sums of each of these metrics across all were computed to represent the total downstream impact of *rworkflows*. All dependents were visualised as nodes in a directed acyclic graph, connecting to an additional node representing the *rworkflows* action (Fig. 2).

To identify the R packages with the highest potential for downstream impact on other packages, we collected data on the number of downloads for every package in CRAN and Bioc using *echogithub*[45]. We then selected the packages with the greatest numbers of downloads and prioritised them for making Pull Requests on their respective GitHub repos to implement *rworkflows*.

An R markdown script to fully reproduce these analyses, as well as an interactive version of the graph with additional metadata, is available as a vignette on the official *rworkflows* GitHub Pages documentation website (See the Code availability section for link).

## GitHub as a package distributor

To comprehensively assess which repositories R packages are distributed via, we collected metadata on all known R packages from base R, CRAN, Bioc, rOpenSci, R-Forge, and GitHub using the package *echogithub*[45]. The total and intersection between packages in each of these repositories were then computed and visualised using the R package *UpSetR*[46] (Fig. 3).

It should be noted that the data on GitHub-hosted R packages comes from a static snapshot previously collected in February 2018 via

the *echogithub* dependency *githubinstall*[47], whereas all the CRAN/Bioc/rOpenSci/R-Forge data is fully up-to-date. This means that our estimates of the proportion of R packages that are distributed exclusively through GitHub are almost certainly an underestimate. An R markdown script to fully reproduce these analyses is available as a vignette on the *rworkflows* documentation website (See the Code availability section).

## Statistics and reproducibility
For the *rworkflows* adoption analysis the total number of GitHub stars, forks, and downloads were summed across all first- and second-order dependents of *rworkflows*. All analysis code can be found on GitHub.

## Reporting summary
Further information on research design is available in the Nature Portfolio Reporting Summary linked to this article.

# Data availability
All code and data to reproduce the analyses performed in this study is shared publicly on GitHub. The latest release of *rworkflows* has been assigned the following Zenodo.

# Code availability
Each component of *rworkflows* is freely available on GitHub[15]. *templateR* R package template: https://github.com/neurogenomics/templateR *rworkflows* R package: https://github.com/neurogenomics/rworkflows *rworkflows* GitHub Action: https://github.com/marketplace/actions/rworkflows *rworkflows* Docker container: https://github.com/neurogenomics/rworkflows/pkgs/container/rworkflows rworkflows Bioconductor vignette: https://neurogenomics.github.io/rworkflows/articles/bioconductor.html *rworkflows* Docker/Singularity container vignette: https://neurogenomics.github.io/rworkflows/articles/docker *rworkflows* dependency graph vignette: https://neurogenomics.github.io/rworkflows/articles/depgraph R package repository distribution vignette: https://neurogenomics.github.io/rworkflows/articles/repos.

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

## Acknowledgements

We would like to thank the development/maintenance teams at GitHub, Bioconductor, and CRAN, as well as the respective contributors of the GitHub Actions that *rworkflows* depends on. We would also like to thank key early adopters of rworkflows for their invaluable feedback and direct contributions; in particular Jacques Serizay, Ali Sajid Imami, and Vince Carey. For the purpose of open access, we have applied a creative commons attribution (CC BY) licence (where permitted by UKRI, 'open government licence' or 'creative commons attribution no-derivatives (CC BY-ND) licence' may be stated instead) to any author accepted manuscript version arising. This work was supported by a UK Dementia Research Institute (UK DRI) Future Leaders Fellowship [MR/T04327X/1] and the UK DRI which receives its funding from UK DRI Ltd., funded by the UK Medical Research Council, Alzheimer's Society and Alzheimer's Research UK. MR/T04327X/1; NGS

## Author contributions

B.M.S. wrote the software described herein, with contributions from A.E.M. B.M.S. wrote the manuscript, with contributions from A.E.M. N.G.S. provided general project guidance.

## Competing interests

The authors declare no competing interests.
