## [Peer Review File · Nature Communications]

rworkflows: automating reproducible practices for the R
communityReviewer #1 (Remarks to the Author):

Schilder *et al.* present a very nicely written article about their suite of tools, *rworkflows*, that promises to improve reproducibility of R packages across different operating systems by using GitHub actions, continuous integration, and containerization. As such, the R community will highly benefit from such a tool. Moreover, the authors already demonstrate that their tool found application in various packages.

Major comments

The tool will be very useful for novel users and package developers, most likely in labs that have no or little R experiences. Therefore, I would suggest that the authors also provide a video tutorial, e.g., on YouTube, or as supplemental file to the article that demonstrates the process of setting up an R package with *rworkflows*, running the continuous integration, and taking a containerized version of the package in Rstudio or on the cluster, and ideally also how to troubleshoot a continuous integration feedback. This will considerably lower the entry barrier for any user and make their tool much more appealing.

How do the authors envision maintenance of *rworkflows* over the coming years as, e.g., R versions or GithubActions might change? I suggest discussing this aspect in the manuscript for such an important suite of tools.

Does *rworkflows* also work on private repositories? Are there strategies to apply continuous integration while keeping a repository private?

Minor comments

- „28-70% of bioinformatics software were never used beyond the original publication“. This seems to be a wide range of percentage. Could the authors verify it (also in light of the 68% percentage mentioned afterwards).
- It might be worth mentioning Zenodo in the introduction as code receives a citable DOI with Zenodo.
- Line 144, "which" should be probably "where"
- Line 151. Could this static snapshot be updated to 2023?
- How was the 91% code coverage in unit tests determined? (line 187)

Reviewer #2 (Remarks to the Author):

The authors addressed a current gap in the development and maintenance of software tools written in the R programming language. Specifically, they developed "*rworkflows*", a robust, reusable, flexible and automated continuous integration (CI) tool for R packages. The authors have tested "*rworkflows*" for over 45 widely downloaded packages on GitHub.

Software tools distributed on GitHub often times don't readily work out-of-the-box. These problems are commonly reported in various online forums, seeking advice on how to resolve various software dependencies and environment requirements. The authors contributed a technically sound method, with a comprehensive GitHub repository and documentation to address a common problem for bioinformaticians. It is a good idea to leverage GitHub Actions for seamless continuous integration.

This manuscript addresses a very important problem and includes statistics on the large number of commonly used bioinformatics R packages distributed on GitHub, thus this manuscript has the potential for high impact. However, the manuscript is written for a technical audience with expertise in software development, and lacks specific use cases to showcase impact. For example, the authors mentioned high impact R packages such as Seurat, RSamtools, GenomeRanges etc. It would be nice to take the readers through some examples of how "*rworkflows*" have improved the usability of high impact packages and analyses of high-throughput data, especially packages that

don't readily work from their GitHub distributions.

Reviewer #3 (Remarks to the Author):

Dear Nature Communications Editors,

I have reviewed with interest the manuscript, "rworkflows: taming the Wild West of R packages." The topic of reproducibility in the R ecosystem is close to my interests and I applaud the authors objectives and implementation in the 'rworkflows' package described here-in. Given this I wish I could be more positive in my overall assessment. However, I am afraid the manuscript has two significant issues: first, it appears to overlook widely used prior work in this area, and second, seems to miss-characterize the root causes of the reproducibility challenges motivating the paper.

The approach taken by the rworkflows package -- facilitating the automatic creation of GitHub Actions scripts for continuous integration -- is commendable. However, the authors make no mention of what I believe is surely the dominant package for providing this precise approach, the package "usethis", (developed by RStudio Inc / Posit Software since 2017 and widely used -- e.g. 254K+ downloads/month, over 145 contributors noted on GitHub). Curiously, the authors compare their effort to the far more niche package maintained by a pair of biological researchers, biocthis, which appears to provide a more Bioconductor-focused adaption of professionally developed usethis. There are some differences between the design and scope of "rworkflows" and "usethis", some of which may be improvements to "usethis" actions, and some other choices that may prove less practical. In an ideal development cycle, the authors would discuss these ideas with the community (e.g. in the openly visible issues tracker on usethis) and propose a pull request -- academic journal review process is a poor substitute for such community review. In this way, these innovations would better benefit from the input of developer community, and if merged into a such a widely used package, would almost surely reach more users than creating an isolated effort. (It is unfortunate that academic incentive structures rarely reward such approaches however.)

My second and more significant concern is in how the paper is motivated. I am very sympathetic to the larger issues of reproducibility regarding computational elements of research, but I found the author's treatment of these issues overly simple and at times misleading.

The underlying tenant of the paper is that a technical solution, in this case, "rworkflows", will have a meaningful impact on researcher behavior (making computational results more reproducible), without any discussion of incentives. This issue is further confused by author's tendency to conflate software written (not necessarily by researchers or academics) with the explicit intent that many others will use it (i.e. software posted to CRAN), with code written to support an analysis for a scientific paper (where the typical assumption is only that it is sufficient documentation of what was done in a single study). For instance, software distributed as such must take measures to run in a landscape of evolving dependencies, require documentation making it accessible and user friendly, testing to ensure it throws appropriate errors when used incorrectly -- none of which are expectations of the script a researcher wrote to run a linear model or create a graph.

The opening claim and title characterize software landscape as a "Wild West" on the basis of the claim that ">50% of R packages are distributed exclusively on GitHub," where there are no minimum standards for submission, rather than CRAN, which (these days) requires some level of standard testing. First, this is a logical fallacy -- the fact that it is possible to put software on GitHub regardless of quality does not demonstrate the quality is poor. In fact, the authors appear to blame the mere existence of public code repositories that lack strict standards: "This has contributed to the scientific landscape becoming the "Wild West" in terms of code usability and reproducibility." Do the authors really believe that the greater transparency created by more research code being posted to public repositories has created this "wild west" rather than just revealing what research code looks like? This perspective in stark contrast to Barnes 2010 Nature

perspective, "Publish your code, it's good enough," which persuasively argues for the need for greater code transparency.

Reviewer #1 (Remarks to the Author):

- *Schilder et al. present a very nicely written article about their suite of tools, rworkflows, that promises to improve reproducibility of R packages across different operating systems by using GitHub actions, continuous integration, and containerization. As such, the R community will highly benefit from such a tool. Moreover, the authors already demonstrate that their tool found application in various packages.*

Major comments

- *The tool will be very useful for novel users and package developers, most likely in labs that have no or little R experiences. Therefore, I would suggest that the authors also provide a video tutorial, e.g., on YouTube, or as supplemental file to the article that demonstrates the process of setting up an R package with rworkflows, running the continuous integration, and taking a containerized version of the package in Rstudio or on the cluster, and ideally also how to troubleshoot a continuous integration feedback. This will considerably lower the entry barrier for any user and make their tool much more appealing.*
 - We thank the reviewer for this excellent suggestion that will be sure to increase usability. We've since uploaded a series of video tutorials to our lab Youtube channel, and embedded links to these tutorials within the README of the main *rworkflows* and *templateR* GitHub repositories. We hope these will help to widen the appeal of *rworkflows* as we begin to more directly advertise it via social media (Twitter/X, LinkedIn).
 - Talk on *rworkflows*: <https://youtu.be/nLIG2prEmCg>
 - Step-by-step vignette on how to use *rworkflows*: <https://youtu.be/vcpMsil3EAU>
- *How do the authors envision maintenance of rworkflows over the coming years as, e.g., R versions or GithubActions might change? I suggest discussing this aspect in the manuscript for such an important suite of tools.*
 - This is an essential question for all research software, especially within academia where personnel turnover is frequent and funding for tool maintenance is usually limited at best. As the first author is a founding member of the official Bioconductor Cloud Methods Working Group, and the primary lead on the GitHub Actions Subgroup, he has already begun discussing this exact concern with the Bioconductor core team via monthly meetings. In addition, he continued this discussion at the Bioc2023 conference (August 2-4, 2023). Alongside other members of the Cloud Methods Working Group, he helped host a Birds of a Feather session to gather feedback from the wider bioinformatics community and determine their needs and preferences. While the specifics are still being worked out, one avenue might be to put in place long-term support from within Bioconductor to ensure that *rworkflows* continues to be maintained in a timely and sustainable manner. In the meantime, the authors of this manuscript

intend to take full responsibility for the maintenance of the *rworkflows* suite. We have strong incentive to do so both for the sake of the community that already utilises *rworkflows*, and for ourselves as all of our lab's R packages now directly rely on it.

- As an additional possible avenue, we have recently proposed to the core RStudio/Posit team that *rworkflows* action be integrated directly into *usethis* (<https://github.com/r-lib/usethis/issues/1880>). We hope this may also open the way for a discussion of the RStudio/Posit team contributing to the long-term maintenance and improvement of the *rworkflows* action. We would ideally like to make *rworkflows* a *de facto* standard for all R users, and then (hopefully soon) elevate it to the level of an officially sanctioned standard.
- *Does rworkflows also work on private repositories? Are there strategies to apply continuous integration while keeping a repository private?*
 - Yes indeed! *rworkflows* can be used regardless of whether the GitHub repository is private or public (assuming the user has the appropriate read/write permissions). This is possible because *rworkflows* requires that a GitHub Token be provided as a GitHub Secret variable. So as long as the token has the necessary permissions, no additional steps need to be taken on the side of the user.
 - We have added this point of clarification to the manuscript (end of the 2nd paragraph of the Discussion):
 - “Furthermore, *rworkflows* can be used in either public or private repositories, extending its utility to pre-production or intellectual property-sensitive packages.”

Minor comments

- „28-70% of bioinformatics software were never used beyond the original publication“. This seems to be a wide range of percentage. Could the authors verify it (also in light of the 68% percentage mentioned afterwards).

- We agree this is an extremely broad range. This partly stems from the fact that we are reporting results from multiple studies with different methodological approaches and phrasing of the results. It is also related to whether the publication introducing the tool counts as a mention/citation towards that tool. Upon re-reading the articles, we believe the range 68-70% is a more accurate comparison of figures reported across studies. We have updated the main text to reflect this:
 - “Despite this, surveys of PubMed and GitHub have revealed that between 68-70% of bioinformatics resources were never used beyond the original publication^{2,3}. “

- It might be worth mentioning Zenodo in the introduction as code receives a citable DOI with Zenodo.

- We thank the reviewer for this suggestion. We have added this to the Introduction (3rd paragraph) with links to the relevant documentation provided in the Supplementary Materials:
 - “There has been extensive integration of GitHub with other resources such as Zenodo and Figshare, enabling the assignment of persistent DOIs with public source code (see **Supplementary Materials: Links** for more details).”

- Line 144, “which” should be probably “where”

- Thanks for the catch, we have amended this as follows:
 - “To evaluate the magnitude of need for GitHub-based solutions in the R community, we gathered comprehensive data on where repositories R packages are hosted.”

- Line 151. *Could this static snapshot be updated to 2023?*

- Unfortunately, the Gepuro Task Views scraper that originally curated the GitHub data has since broken was abandoned by the developers *circa* 2018. See here for details:
 - <https://github.com/hoxo-m/githubinstall/issues/41>
 - <https://github.com/hoxo-m/gepuro-task-views-copy/issues/1>
- To address this, we are working on implementing our own scraping function within the *echogithub* package. Presently, GitHub’s regex-enabled code search is not API-accessible, making this difficult to extract comprehensively. However, based on our initial search we can see that there are upwards of 89.6k R packages on GitHub (see link below). However, this does not account for forks and thus programmatically gathering these DESCRIPTION files and computing the unique number of packages will be necessary to say exactly how many there are.
 - <https://github.com/search?q=%2F%28%3F-i%29Package%3A%2F+path%3A%2F%28%3F-i%29%5EDESCRIPTION%24%2F&type=code>
- We have reached out to the GitHub team requesting API access to the advanced search, and will continue to look for ways to address this in the meantime. We aim to have this updated data ready in time for the final publication of this manuscript.

- *How was the 91% code coverage in unit tests determined? (line 187)*

- We have added a section to the Discussion explaining more thoroughly the meaning and utility of code coverage tests:
 - “Code coverage is one particularly useful metric for assessing package robustness. While the precise measurement of code coverage varies slightly from one implementation to another, it can generally be summarised as the proportion of lines in your software’s code that are run during unit tests (e.g. using the *testthat* framework). This takes into account that code within conditional statements may not be run in all scenarios, and thus encourages developers to test the same code using multiple sets of parameters. Assuming

that the tests themselves are valid, a code coverage of 91% could be interpreted as “91% of its code has been systematically tested and is working as intended”. Thus, code coverage can serve as a useful, continuous measure of package robustness as it reduces the chances that the code will break in various use cases^{26–29}. The *rworkflows* action automatically runs code coverage tests via *covr* (Hester) and uploads a report to the browser-based Codecov or Coverall services where users can interactively explore which portions of their code are currently not being thoroughly tested. Finally, the *rworkflows::use_badges* function (which builds upon the *badger* package) allows developers to easily advertise both discrete metrics (passing on CRAN/Bioc, passing *rworkflows* checks) and continuous metrics (code coverage percentage) on their GitHub landing pages.”

Reviewer #2 (Remarks to the Author):

The authors addressed a current gap in the development and maintenance of software tools written in the R programming language. Specifically, they developed "rworkflows", a robust, reusable, flexible and automated continuous integration (CI) tool for R packages. The authors have tested "rworkflows" for over 45 widely downloaded packages on GitHub.

Software tools distributed on GitHub often times don't readily work out-of-the-box. These problems are commonly reported in various online forums, seeking advise on how to resolve various software dependencies and environment requirements. The authors contributed a technically sound method, with a comprehensive GitHub repository and documentation to address a common problem for bioinformaticians. It is a good idea to leverage GitHub Actions for seamless continuous integration.

This manuscript addresses a very important problem and includes statistics on the large number of commonly used bioinformatics R packages distributed on GitHub, thus this manuscript has the potential for high impact. However, the manuscript is written for a technical audience with expertise in software development, and lacks specific use cases to showcase impact. For example, the authors mentioned high impact R packages such as Seurat, RSamtools, GenomeRanges etc. It would be nice to take the readers through some examples of how "rworkflows" have improved the usability of high impact packages and analyses of high-throughput data, especially packages that don't readily work from their GitHub distributions.

- We thank the reviewer for their insightful comments. *rworkflows* has already been used to reveal bugs in both Rsamtools (<https://github.com/Bioconductor/Rsamtools/pull/48>) and GenomicRanges (<https://github.com/Bioconductor/GenomicRanges/pull/73>) that were not apparent until after the developers had pushed to Bioconductor. Had *rworkflows* been used, the developers would have been made aware of these bugs (within 15-30 minutes) before pushing upstream to Bioconductor, and thus not propagate the bugs to all users.

- While we could use these as examples, we feel it wouldn't be appropriate to single out other packages/maintainers within this manuscript. Instead, we've opted to use one of our lab's own packages as an example, under the new Results subsection "Use case: MAGMA.Celltyping".
- In addition, we have recorded a series of Youtube videos walking users step-by-step through how to use `rworkflows`. We will continue updating and adding to these videos over time.
 - https://youtube.com/playlist?list=PL4pSvJm1oWAvlKx1f8W4AZFkKN_Ev65fb&si=gq6SS2D8ZIFNNUj

Reviewer #3 (Remarks to the Author):

Dear Nature Communications Editors,

I have reviewed with interest the manuscript, "rworkflows: taming the Wild West of R packages." The topic of reproducibility in the R ecosystem is close to my interests and I applaud the authors objectives and implementation in the 'rworkflows' package described here-in. Given this I wish I could be more positive in my overall assessment. However, I am afraid the manuscript has two significant issues: first, it appears to overlook widely used prior work in this area, and second, seems to miss-characterize the root causes of the reproducibility challenges motivating the paper.

The approach taken by the rworkflows package -- facilitating the automatic creation of GitHub Actions scripts for continuous integration -- is commendable. However, the authors make no mention of what I believe is surely the dominant package for providing this precise approach, the package "usethis", (developed by RStudio Inc / Posit Software since 2017 and widely used -- e.g 254K+ downloads/month, over 145 contributors noted on GitHub). Curiously, the authors compare their effort to the far more niche package maintained by a pair of biological researchers, biocthis, which appears to provide a more Bioconductor-focused adaption of professionally developed usethis.

- We thank the reviewer for their insightful and relevant comments. At the end of the Introduction we include the link to a step-by-step tutorial on making an R package with `usethis` + `biocthis` and/or `rworkflows` (which details the differences between these approaches at each step). However, we failed to mention `usethis` within the manuscript which is indeed an oversight in retrospect. We have amended this by expanding the Results to include comparisons with both packages, now entitled "Comparisons with `usethis`/`biocthis`".
- Another unfortunate byproduct of workflow-based implementations like `usethis`/`biocthis` is that there is currently no way to record usage stats. As popular as `usethis` is, it should be noted that only a small proportion of its functions are related to GHA workflows. This makes it difficult to quantify how many users are utilising the GHA workflows-related functions (i.e. the `usethis::use_github_action*` series of functions), as opposed to all the other very useful non-GHA related functions. While these features may very well be used by a number of developers, we are unaware of any way it would be possible to make definitive statements about the extent to which this is true. This is not the case for action-based implementations like `rworkflows`,

where the full list of repos that use the action can be found under the Insights tab (which is how we acquired the data for Fig 2).

There are some differences between the design and scope of "rworkflows" and "usethis", some of which may be improvements to "usethis" actions, and some other choices that may prove less practical.

- We would once more like to clarify that *usethis* does not currently have any dedicated GitHub actions (to our knowledge), only static workflow scripts.
- Nevertheless, one possibility that arises from *rworkflows* being GH action-based is that it can be easily called from other actions and workflow scripts (in a version-controlled manner). As such, we have proposed to the core Rstudio/Posit team that *rworkflows* action be integrated directly into the *usethis*-generated workflow scripts (<https://github.com/r-lib/usethis/issues/1880>). The hope is that this may open new avenues for the professional long-term maintenance and improvement of the *rworkflows* suite.

In an ideal development cycle, the authors would discuss these ideas with the community (e.g. in the openly visible issues tracker on usethis) and propose a pull request -- academic journal review process is a poor substitute for such community review. In this way, these innovations would better benefit from the input of developer community, and if merged into a such a widely used package, would almost surely reach more users than creating an isolated effort. (It is unfortunate that academic incentive structures rarely reward such approaches however.)

- We openly invite all users to report bugs, request features, and provide PRs via the dedicated *rworkflows* Github repository (the same mechanism that *usethis* employs). We have already received detailed feedback from the creators of *biocthis* (<https://github.com/neurogenomics/rworkflows/issues/21>), and merged several useful PRs from users (<https://github.com/neurogenomics/rworkflows/pulls?q=>). That said, the *rworkflows* suite would certainly benefit from greater engagement with the R community. We feel it is of the utmost importance to work with and serve the needs of the R community, not impose arbitrary standards which make their work more difficult.

My second and more significant concern is in how the paper is motivated. I am very sympathetic to the larger issues of reproducibility regarding computational elements of research, but I found the author's treatment of these issues overly simple and at times misleading.

The underlying tenant of the paper is that a technical solution, in this case, "rworkflows", will have a meaningful impact on researcher behavior (making computational results more reproducible), without any discussion of incentives.

- We agree that making clear the incentives for multiple stakeholders in the R community is essential to the success of any infrastructure-based project. It is for this reason we previously dedicated much of the Discussion to explicitly listing these incentives. To recapitulate, these include:

- a. Reducing the burden on developers to implement robust and reproducible software. This also makes it easier for developers to return to projects after long periods of inactivity, due to insufficient documentation or versioning conflicts.
 - b. Providing journal editors and reviewers with a means of assessing package quality. Conversely, authors can provide the successful `rworkflows` checks and code coverage as a way to easily communicate the robustness of their R package (e.g. “`rworkflows` has 91% test coverage”).
 - c. Bypassing the vast majority of installation issues that users commonly face via the simplified creation and distribution of Docker/Singularity containers. This is also useful to the developer, as they will be less burdened by help requests for common installation issues.
 - d. Providing a systematic methodology and set of standards for research groups to ensure their will is sufficiently documented and reproducible even after they have left the group.
 - e. More usable software gets cited more (see first paragraph of Introduction).
- We have further expanded on these points in the Discussion for the sake of clarity. If the reviewer feels we have omitted any other important sources of incentivization, we would be highly amenable to including those as well.

This issue is further confused by author's tendency to conflate software written (not necessarily by researchers or academics) with the explicit intent that many others will use it (i.e. software posted to CRAN), with code written to support an analysis for a scientific paper (where the typical assumption is only that it is sufficient documentation of what was done in a single study). For instance, software distributed as such must take measures to run in a landscape of evolving dependencies, require documentation making it accessible and user friendly, testing to ensure it throws appropriate errors when used incorrectly -- none of which are expectations of the script a researcher wrote to run a linear model or create a graph.

- The reviewer has raised an important point regarding our language when referring to R package developers. As researchers ourselves, we sometimes forget to use more inclusive terms such as “developer” vs. “researcher”. We have taken care to address this throughout the manuscript to make clear that `rworkflows` is intended for all developers of R packages, not just those within research settings.
- We certainly agree that distributing code from a given project as a fully developed piece of software is not always appropriate or necessary. It is for this reason that `rworkflows` is only designed for use with R packages, not all code. That said, we do believe there is significant room for improvement when it comes to reproducibility in publication-associated code more generally, even unpackaged scripts. While we cannot offer a simple solution to this broader issue here, we do believe there are some general principles (FAIR and otherwise) central to `rworkflows` that can be applied to code more generally (e.g. documentation, containerisation). For example, if the package to run the linear model (or its unstated dependencies) cannot be identified from the script, or the steps taken to preprocess the data to feed into the model are

untraceable, this is problematic for the state of published work. Despite not needing to be a package, these issues would limit its utility for the broader community and make the validity of the overall work more difficult to assess.

The opening claim and title characterize software landscape as a "Wild West" on the basis of the claim that ">50% of R packages are distributed exclusively on GitHub," where there are no minimum standards for submission, rather than CRAN, which (these days) requires some level of standard testing. First, this is a logical fallacy -- the fact that it is possible to put software on GitHub regardless of quality does not demonstrate the quality is poor. In fact, the authors appear to blame the mere existence of public code repositories that lack strict standards: "This has contributed to the scientific landscape becoming the "Wild West" in terms of code usability and reproducibility." Do the authors really believe that the greater transparency created by more research code being posted to public repositories has created this "wild west" rather than just revealing what research code looks like?

- Our assessment of the current landscape of R package development was in no way meant to blame or offend R package developers. Rather, it meant simply to reflect the present situation we find ourselves in as a community. We ourselves have produced (and continue to produce) a number of open-source R packages, many of which are currently exclusively distributed via GitHub. In fact, the creation of *rworkflows* was one born out of necessity to assess the quality of our own software and make it easier to use/maintain in the long run.
- We have added a paragraph to the beginning of our Discussion that we feel summarises our view on this topic:
 - Much like the Wild West, the landscape of R packages includes "the good, the bad, and the ugly". There are no doubt hundreds of invaluable and high-quality R packages that are not hosted on CRAN/Bioc/rOpenSci. Nevertheless, distinguishing the "good" from the "bad" from the "ugly" is currently a process of trial and error, which amounts to a huge number of wasted hours compounded across thousands of users. *rworkflows* attempts to make these distinctions more visible and immediately accessible, all while helping to make all R packages meet a set of minimum standards (or at least transparently advertise that they don't yet).
- In addition, we offer one of our own packages as an example of how *rworkflows* can be used to improve quality and maintainability, in new section titled "Use case: MAGMA.Celltyping".

This perspective in stark contrast to Barnes 2010 Nature perspective, "Publish your code, it's good enough," which persuasively argues for the need for greater code transparency.

- We found this article interesting and informative. We also respectfully disagree on the reviewer's interpretation of it. Publishing the code is a fantastic and absolutely necessary first step of any study, regardless of what state it is in. But if we're to effectively make advances as a community, we need to aim to do better. Providing tools to make reproducible practices as easy and incentivised as possible is our group's attempt to move towards that goal.

- We would also like to point out that this is relevant to the topic of transparency, in the sense that if an article describing a novel software is published in a peer reviewed journal, it should work as advertised. Unfortunately, there are many R packages and software more generally that fail to meet this criterion, even at the time of initial publication. We find this very problematic for the state of the published literature, as it misleads and erodes trust in the mechanisms that are intended to safeguard against these scenarios. This is not to suggest that such failures are necessarily intentional on the part of the developers (they may be unaware of their work's shortcomings). Therefore, we would argue that publishers, authors, and readers alike should be naturally incentivised to put in place some guard rails for the sake of improving the quality of the work and advancing the field.

Reviewer #1 (Remarks to the Author):

The authors have addressed my comments and I recommend the article for publication. I hope that rworkflows will find wide applications and am looking forward to seeing more video tutorials coming up on Youtube.

Reviewer #3 (Remarks to the Author):

Dear authors,

Thank you for these detailed revisions and an excellent piece. I think your work is exemplary, and (or because) I think the issues you raise here are very important, please forgive me if I drag this out a little longer.

I still have some core questions after reading the revisions that to me are not answered.

- Will a significant fraction of researchers ever adopt this approach?
- What does it take to maintain the ecosystem to do this?

Anonymous peer review is a funny thing, and I appreciate that from a few short paragraphs it is hard to see where my comments are coming from or what the context is behind them. So in the chance that it is at all helpful in better understanding what I'm trying to say here, let me try and give a bit of my context before going into specifics. I co-founded the rOpenSci project, I co-founded the Rocker project (which provides the base docker images used by bioconductor team which you in turn use as a base image in rworkflows), and have maintained these along with numerous R packages on CRAN for over a decade. I have used the R package structure to package code on GitHub for dozens of papers not destined for CRAN, following what Gentleman & Lang (both contributors to R-Core) dubbed the "compendium" approach in 2007 (<https://www.jstor.org/stable/27594227>; i.e. that the "package" mechanism helps make code more reproducible even when that code simply reproduces the results of a paper and isn't intended to be the more general-purpose software of the type found on CRAN). I've authored my own version of papers calling for more authors to adopt that approach (2017, <https://doi.org/10.1080/00031305.2017.1375986>), in which we emphasized newer tools such as Docker as well. I mention these things not to toot my own horn here, but only because I know just how you feel when I ask, "Will a significant fraction of researchers ever adopt this approach."

Believe me, I applaud your spirit in answering Nick Barnes 2010 piece with -- "no, it's really not enough", and personally I couldn't agree more with you. But at that time, simply getting researchers code to ever see the light of day anywhere was a *huge step forward* in reproducibility. And while the situation is better today -- We now have a whole "Wild West" of researcher code instead of papers that read "code available on request" -- the practice is far from universal. Barnes' title was not a rejection of the need for much better software practices, it was an acknowledgement of the reality that setting the bar too high will discourage changes in adoption. I do not see that nuance reflected in the rworkflows paper -- it focuses only on the technical side, seemingly from the angle of 'set the bar as high as possible', and argue that 36 examples (& how many unique authors) is progress. Yes it's a high bar -- I know from helping run the software review process we created at rOpenSci that has seen 100s of individual authors learn how to cross a similar threshold. It's not high just because it's hard to create all the actions infrastructure (though rworkflows does a nice job here, as do other tools), but high because writing unit tests and documentation and debugging take a lot of time. As the authors acknowledge, CRAN is also a reasonably high bar (if not always in the best of ways). But the point of this paper, as I understand it, is to offer a somewhat lower bar than these established mechanisms for widespread distribution, a mechanism that is meant to be more accessible. Gentleman & Lang explicitly used the 'compendium' term to suggest a lighter-weight and therefore more easily crossed threshold that recognized the compromises a researcher team could make to provide reproducible code that simply didn't need the same level of unit tests, documentation, and other best practices expected of general purpose software on CRAN that is meant to be

downloaded and re-used in contexts far from reproducing a paper. It feels discordant to me that the here, the rworkflows paper doesn't wrestle with or explain the authors views on this trade-off: is it your view that any code accompanying a paper should meet these checks? I know rworkflows checks have various opt-in steps -- not everything needs to pass bioc check, say, but to me this misses smaller but more reasonable actions that would be a lower bar than passing automated packaging checks, such as dynamic rendering of an Rmd or quarto doc in GitHub Actions (as is already provided by, say, quarto, or the Notebooks Now Initiative).

On to "what does this take to maintain?" To pick just one concrete example, setting up system requirements as software evolves is a challenge that requires active maintenance. For instance, the rworkflows action that handles this on R is actually pulling the Ubuntu 20.04 (focal), (e.g. <https://github.com/neurogenomics/rworkflows/blob/29a383416fc3b0c26ebb6813de2380dbdea3c297/action.yml#L154>) though Ubuntu latest is now on 22.04 (jammy)). Most but not all package names are the same between these two releases (especially for dev libs), but this will break in some cases, and will need to be actively maintained going forward as well. Beyond dependencies, new architectures emerge like M1 on Apple's macs, and the subsequent need to develop actions that support that architecture, and not just the three operating systems on a single architecture currently. More to the point, this functionality provided in this step, as in many of the steps in rworkflows (pkgdown, vignettes, test, rcmdcheck, and others), are actively maintained by both a large professional and volunteer community in r-lib/actions (<https://github.com/r-lib/actions/tree/v2-branch/examples>). There are definitely novel contributions here, and I don't raise this to disparage the effort, only that these are non-trivial issues that would give me pause in depending on these actions, just as I would in evaluating my choice of software dependencies in any other context. These are complex problems that are best addressed by a larger community effort, and only then with continued input, a defined governance model, and so forth.

So, where to? I think my first concern is easily addressed by a mention in the intro and discussion to acknowledge that it is not sufficient to simply argue "we can and should do better." Sure -- every paper could be accompanied by a stellar package on CRAN or rOpenSci or BioC too -- but the point here is about feasible stepping stones, not pie-in-the-sky. We could quibble over what those stepping stones are, but the idea is there in rworkflows, just acknowledge that researchers feel this tension, and how rworkflows provides stepping stones and not an all-or-nothing approach.

On maintenance, I'm just making those comments as future-looking ideas to make rworkflows easier for you to maintain for years to come and attract more users. The paper can briefly mention those plans, or not, rworkflows will evolve from whatever you say in the paper anyway as good software always does. I know actions development is a bit of a funny space, where depending on upstream implementations in other actions or scripts isn't as clean as in an actual programming language, but I encourage you to try and inherit more directly from *community maintained* actions, instead of copying over and modifying code that you will have to then keep synchronized. I like this paper, I love your work in this space, and I'd love to see rworkflows succeed.

All the best,

Reviewer #1 (Remarks to the Author):

The authors have addressed my comments and I recommend the article for publication. I hope that rworkflows will find wide applications and am looking forward to seeing more video tutorials coming up on Youtube.

- We thank the reviewer for their insightful comments and suggestions to further improve this manuscript.

Reviewer #3 (Remarks to the Author):

Dear authors,

Thank you for these detailed revisions and an excellent piece. I think your work is exemplary, and (or because) I think the issues you raise here are very important, please forgive me if I drag this out a little longer. I still have some core questions after reading the revisions that to me are not answered.

- *Will a significant fraction of researchers ever adopt this approach?*
- *What does it take to maintain the ecosystem to do this?*

*Anonymous peer review is a funny thing, and I appreciate that from a few short paragraphs it is hard to see where my comments are coming from or what the context is behind them. So in the chance that it is at all helpful in better understanding what I'm trying to say here, let me try and give a bit of my context before going into specifics. I co-founded the rOpenSci project, I co-founded the Rocker project (which provides the base docker images used by bioconductor team which you in turn use as a base image in rworkflows), and have maintained these along with numerous R packages on CRAN for over a decade. I have used the R package structure to package code on GitHub for dozens of papers not destined for CRAN, following what Gentleman & Lang (both contributors to R-Core) dubbed the "compendium" approach in 2007 (<https://www.jstor.org/stable/27594227>; i.e. that the "package" mechanism helps make code more reproducible even when that code simply reproduces the results of a paper and isn't intended to be the more general-purpose software of the type found on CRAN). I've authored my own version of papers calling for more authors to adopt that approach (2017, <https://doi.org/10.1080/00031305.2017.1375986>), in which we emphasized newer tools such as Docker as well. I mention these things not to toot my own horn here, but only because I know just how you feel when I ask, "Will a significant fraction of researchers ever adopt this approach." Believe me, I applaud your spirit in answering Nick Barnes 2010 piece with -- "no, it's really not enough", and personally I couldn't agree more with you. But at that time, simply getting researchers code to ever see the light of day anywhere was a *huge step forward* in reproducibility. And while the situation is better today -- We now have a whole "Wild West" of researcher code instead of papers that read "code available on request" -- the practice is far from universal. Barnes' title was not a rejection of the need for much better software practices, it was an acknowledgement of the reality that setting the bar too high*

will discourage changes in adoption. I do not see that nuance reflected in the rworkflows paper -- it focuses only on the technical side, seemingly from the angle of 'set the bar as high as possible', and argue that 36 examples (& how many unique authors) is progress. Yes it's a high bar -- I know from helping run the software review process we created at rOpenSci that has seen 100s of individual authors learn how to cross a similar threshold. It's not high just because it's hard to create all the actions infrastructure (though rworkflows does a nice job here, as do other tools), but high because writing unit tests and documentation and debugging take a lot of time. As the authors acknowledge, CRAN is also a reasonably high bar (if not always in the best of ways). But the point of this paper, as I understand it, is to offer a somewhat lower bar than these established mechanisms for widespread distribution, a mechanism that is meant to be more accessible. Gentleman & Lang explicitly used the 'compendium' term to suggest a lighter-weight and therefore more easily crossed threshold that recognized the compromises a researcher team could make to provide reproducible code that simply didn't need the same level of unit tests, documentation, and other best practices expected of general purpose software on CRAN that is meant to be downloaded and re-used in contexts far from reproducing a paper. It feels discordant to me that here, the rworkflows paper doesn't wrestle with or explain the authors views on this trade-off: is it your view that any code accompanying a paper should meet these checks? I know rworkflows checks have various opt-in steps -- not everything needs to pass bioc check, say, but to me this misses smaller but more reasonable actions that would be a lower bar than passing automated packaging checks, such as dynamic rendering of an Rmd or quarto doc in GitHub Actions (as is already provided by, say, quarto, or the Notebooks Now Initiative).

- First of all, we thank Dr. Boettiger for his open and honest discussion of their background and experience as it relates to reproducibility in R and computational research more generally. In light of the reviewer's insightful feedback, we have attempted to modify the overall tone of the paper to be less punitive and regulatory, to more inclusive and helpful. For example, the title has been changed from "*rworkflows: taming the Wild West of R packages*" to "*rworkflows: automating reproducible practices for the R community*".
- The abstract has similarly been rewritten to emphasize the importance of sharing code on GitHub, as well as key parts of the manuscript body. We hope these changes will both better capture the spirit of what we are trying to achieve, and promote adoption rather than intimidation.
- While it still remains to be seen exactly how widespread *rworkflows* will become, we are hopeful it will be positively received by the R developer community. This is partly based on community feedback at several talks and workshops presented by the first author. Since the last review the number of repositories using *rworkflows* has already grown to >100. Publishing this paper is one step towards lending our approach credibility and broader awareness. We continue to aim to make it as easy and incentivized as possible for researchers to use *rworkflows*.

On to "what does this take to maintain?" To pick just one concrete example, setting up system requirements as software evolves is a challenge that requires active maintenance. For instance, the rworkflows action that handles this on R is actually pulling the Ubuntu 20.04 (focal), (e.g.

<https://github.com/neurogenomics/rworkflows/blob/29a383416fc3b0c26ebb6813de2380dbdea3c297/a>

ction.yml#L154) though Ubuntu latest is now on 22.04 (jammy)). Most but not all package names are the same between these two releases (especially for dev libs), but this will break in some cases, and will need to be actively maintained going forward as well.

- This is indeed a valid concern. In fact, this exact issue was recently spotted by a user, and subsequently been resolved by defaulting to the latest release of the RSPM:
<https://github.com/neurogenomics/rworkflows/issues/73>

Beyond dependencies, new architectures emerge like M1 on Apple's macs, and the subsequent need to develop actions that support that architecture, and not just the three operating systems on a single architecture currently. More to the point, this functionality provided in this step, as in many of the steps in rworkflows (pkgdown, vignettes, test, rcmdcheck, and others), are actively maintained by both a large professional and volunteer community in r-lib/actions (<https://github.com/r-lib/actions/tree/v2-branch/examples>). There are definitely novel contributions here, and I don't raise this to disparage the effort, only that these are non-trivial issues that would give me pause in depending on these actions, just as I would in evaluating my choice of software dependencies in any other context. These are complex problems that are best addressed by a larger community effort, and only then with continued input, a defined governance model, and so forth.

- While rworkflows has been stable and running consistently across dozens of unique R package repositories, we agree that long-term maintenance of open-source non-commercial software is a major challenge. Accordingly, we have added a dedicated paragraph addressing this topic to the Discussion:
 - “In that same vein, we recognise the importance of ensuring rworkflows itself is maintained and extended well into the foreseeable future. We are committed to securing this vision, out of a desire to make this resource continually available to the R community as well as our reliance on rworkflows for all of our R packages. It is for this very reason that the rworkflows action was designed to, whenever possible, use subactions from well-established developer organisations (e.g. GitHub, R Consortium, Posit, Bioc) as they have the highest likelihood of being maintained moving forward. In addition, we have already begun to explore multiple avenues towards open-source longevity including (but not limited to) seeking official support/collaboration with software repositories, user contributions, crowd-funding, corporate sponsorship, and grants for sustainable software development. All members of the community are encouraged to voice their ideas/concerns/opinions by participating in the dedicated “Longevity” Discussion board (<https://github.com/neurogenomics/rworkflows/discussions>).”
- More specifically, there are also ongoing discussions within the Bioc core team, Bioc Cloud Methods Working Group, and broader Bioc community, regarding the possibility of transitioning the entire Bioc infrastructure to GitHub. If this were pursued, rworkflows is well-positioned to become a pillar of this new GH-based infrastructure, ensuring it would be continually maintained well into the foreseeable future. While this possibility is far from being solidified, it

does offer a potential additional avenue towards securing long-term support for *rworkflows*. As this is still at an early stage, we don't feel we're at liberty to cite this specific example in the context of the manuscript, we have nevertheless alluded to the possibility of receiving long-term support via the general mechanism of being taken up by large R-focused consortia.

So, where to? I think my first concern is easily addressed by a mention in the intro and discussion to acknowledge that it is not sufficient to simply argue "we can and should do better." Sure -- every paper could be accompanied by a stellar package on CRAN or rOpenSci or BioC too -- but the point here is about feasible stepping stones, not pie-in-the-sky. We could quibble over what those stepping stones are, but the idea is there in rworkflows, just acknowledge that researchers feel this tension, and how rworkflows provides stepping stones and not an all-or-nothing approach.

- We agree wholeheartedly with this sentiment and have tried to revise the manuscript to better appeal to the wide variety of stakeholders that may benefit from *rworkflows* (including developers with different experience levels and project-specific objectives).
- For example, in the final paragraph of the Discussion we have added the following:
 - "Furthermore, the *rworkflows* action is designed to be both easy to use and flexible (through customisable parameters), thus enabling developers to utilise it in whatever way best suits their project-specific goals. In practice, this can range from checking for basic installability/usability of an R package, all the way to extensive evaluation of consortia-specific coding and documentation standards with fully automated container deployment. "

*On maintenance, I'm just making those comments as future-looking ideas to make rworkflows easier for you to maintain for years to come and attract more users. The paper can briefly mention those plans, or not, rworkflows will evolve from whatever you say in the paper anyway as good software always does. I know actions development is a bit of a funny space, where depending on upstream implementations in other actions or scripts isn't as clean as in an actual programming language, but I encourage you to try and inherit more directly from *community maintained* actions, instead of copying over and modifying code that you will have to then keep synchronized.*

- We absolutely agree with this strategy, and indeed, it is the one that *rworkflows* takes wherever possible. All actions that *rworkflows* currently employs is listed at the following URL: <https://github.com/neurogenomics/rworkflows/network/dependencies>
- We have also added a sentence to the Discussion explaining this strategy (see above excerpt.)

*I like this paper, I love your work in this space, and I'd love to see rworkflows succeed.
All the best,*

- Thank you again for all the time and attention to detail you have put into reviewing this work. It has undoubtedly made both the manuscript and the software better, and (with any luck) increased its chances of becoming a foundational tool for the R community.